# Osteoclasts and Probiotics Mediate Significant Expansion, Functional Activation and Supercharging in NK, γδ T, and CD3+ T Cells: Use in Cancer Immunotherapy

**DOI:** 10.3390/cells13030213

**Published:** 2024-01-24

**Authors:** Kawaljit Kaur, Anahid Jewett

**Affiliations:** 1Division of Oral Biology and Medicine, School of Dentistry and Medicine, University of California, Los Angeles, CA 90095, USA; drkawalmann@g.ucla.edu; 2The Jonsson Comprehensive Cancer Center, School of Dentistry and Medicine, University of California, Los Angeles, CA 90095, USA

**Keywords:** osteoclasts, NK cells, supercharged NK cells, CD3+ T cells, γδ T cells, IFN-γ, humanized-BLT mice

## Abstract

Our previous studies have introduced osteoclasts (OCs) as major activators of NK cells. It was found that OCs exhibit the capabilities of inducing cell expansion as well as increasing the cytotoxic activity of NK cells by granule release and increasing the secretion of TNF-α and TRAIL, leading to increased lysis of tumors in short-term as well as long-term periods, respectively. OC- induced expanded NK cells were named supercharged NK cells (sNK) due to their significantly high functional activity as well as their significantly higher cell expansion rate. It is, however, unclear whether the OC-mediated effect in NK cells is specific or whether other cytotoxic immune cells can also be expanded and activated by OCs. We chose to focus on γδ T cells and pan T cells, which also include CD8+ T cells. In this paper, we report that OCs are capable of expanding and functionally activating both γδ T cells and pan T cells. Expanded γδ T and pan T cells were capable of secreting high levels of INF-γ, albeit with different dynamics to those of NK cells, and, moreover, they are unable to kill NK-specific targets. Since we used humanized-BLT (hu-BLT) mice as a model of human disease, we next determined whether NK and T cell activation through OCs is also evident in cells obtained from hu-BLT mice. Similar to humans, OCs were capable of increasing the cell expansion and secretion of IFN-γ in the culture of either NK or T cells from hu-BLT mice, providing yet further evidence that these mice are appropriate models to study human disease. Therefore, these studies indicated that CD3+ T or γδ T cells can proliferate and be supercharged by OCs similar to the NK cells; thus, they can be used individually or in combination in the cell therapy of cancers.

## 1. Introduction

Natural killer (NK) cells constitute 5–10% of total immune cells in human peripheral blood mononuclear cells (PBMCs) [1,2,3]. NK cell-released pre-formed granules, such as granzyme B and perforin, play a significant role in NK cell-induced cytotoxic function by inducing apoptosis and/or programmed cell death in target cells [4,5,6]. NK cells target tumors either by direct NK cell-induced killing or by antibody-dependent cellular cytotoxicity (ADCC). NK cell-secreted cytokines and chemokines play a significant role in the NK cell function of regulating other immune cells [2,3]. NK cells secrete several cytokines, but more important ones are granulocyte macrophage-colony stimulating factor (GM-CSF), interleukin-10 (IL-10), IL-13, interferon-γ (IFN-γ), tumor necrosis factor-α (TNF-α), and TNF-β [3,7,8]. Among these, TNF-α and IFN-γ play a crucial role in inducing differentiation in cancer stem-like cells (CSCs) and/or undifferentiated tumors [9,10,11]. Downmodulation of NK cell numbers and/or function in the peripheral blood of patients were found to be directly correlated with higher cancer risk, cancer metastasis, and poor prognosis of cancer patients [12,13,14,15,16,17,18,19,20,21,22,23]. To date, NK cell-based immunotherapies have gained significant attention because many techniques were demonstrated for in vitro NK cell proliferation, resulting in higher NK cell numbers for therapeutics as well as increased in vivo functional activity and proliferation of NK cells [24,25,26,27,28,29,30,31,32,33,34,35,36,37,38,39]. In addition, NK cells lack graft versus host disease (GVHD), which allows the use of autologous as well allogeneic NK cells for adoptive cell therapies [40,41,42,43].

We have previously demonstrated NK expansion methodology, in which we used osteoclasts (OCs) and sAJ2 sonicated probiotic bacteria to activate NK cells. This methodology resulted in the production of NK cells exhibiting great anti-cancer activity. We later named these highly potent NK cells supercharged NK cells (sNK) [44]. Macrophage colony-stimulating factor (M-CSF) and the receptor activator of nuclear factor-κB ligand (RANKL) stimulate bone marrow- or peripheral blood-derived monocyte/macrophage cells for OC generation [45,46,47,48,49,50]. RANKL expressed on NK and T cells can induce osteoclast genesis during their interaction with monocytes. OCs were found to be potent activators of NK cells when OCs were compared to monocytes, macrophages, dendritic cells, K562, OSCSCs, MP2, and PBMCs [51,52]. In comparison to other feeder cells used in our study, OCs were superior in inducing the cell expansion, increased cytotoxicity, and increased secretion of cytokines or chemokines in NK cells [51,52]. OCs secreted cytokines such as IL-15, IL-12, IL-18 and IFN-α, and the surface phenotype of OCs MHC-class-I^dim^MHC-Class-II^dim^CD14^dim^CD11b^dim^CD54^dim^ play a role in the activation of NK cells. Furthermore, the MHC-class I surface expression level of OCs was not modulated much when treated with supernatants from the activated NK cells (IFN-γ from activated NK cell supernatant results in increased expression of MHC-class I in target cells) [51]. To sum up, the OC-secreted factors and their surface phenotype (lower expression of MHC-class I) could be the underlying mechanisms by which OCs are capable of expanding highly potent NK cells. In addition, OCs are positive for NKG2D ligand; this ligand is known to increase NK function by its binding to the NKG2D receptor in NK cells. Probiotic sAJ2 also contributes to activating and expanding NK cells; one of the mechanisms is increased cytokine secretion in NK cells including IFN-γ. In addition, probiotic bacteria increased the survivability of NK cells [10]. Hence, the combination of probiotics and OCs induced signals that participate additively to induce functional activation and increased proliferation of NK cells. OC and probiotic-expanded sNK cells were found to be highly potent in killing and differentiating tumors both in our in vitro and in vivo cancer studies [53,54,55].

T cells comprise the major portion of PBMCs, and CD8+ T cells were the most focused subset in the field of cancer immunotherapies [56,57,58,59,60,61]. However, recent studies have highlighted the anti-cancer function of γδ T and CD4+ T cells [62,63,64,65,66,67,68,69,70,71,72,73]. In previous studies, we used OCs and sAJ2 probiotic bacteria to induce cell expansion in CD3+ T and CD8+ T cells and observed OC-induced expansion in T cells [74]. This is our first study to show the expansion of γδ T cells using the combination of OCs and probiotic bacteria.

Although many mice are/were used in human pathology and cancer studies, studies have demonstrated differences in human vs. mouse immune systems [75,76]. In this study, we used humanized-BLT (hu-BLT; human bone marrow/liver/thymus) mice which represent an established and most accepted humanized mouse model for human studies [77,78]. Hu-BLT mice generation is achieved in NSG mice by the surgical implantation of human fetal liver and thymus tissue fragments under the kidney capsule, and for supporting the full reconstitution of the human bone marrow, the autologous CD34^+^ hematopoietic stem cells are injected via tail-vein IV injection [79,80]. The use of such mice has demonstrated great success in determining mechanisms of underlying diseases and therapeutic approaches to human diseases [77,78].

In this study, our goal is to delineate the differences and similarities in the functional activation of NK and T cells. We first observed that T cells were incapable of killing NK cell-specific CSCs. In addition, we compared NK, CD3+ T, and γδ T cells during their interaction with OCs and probiotic bacteria sAJ2. Further, we demonstrated the close similarity of NK and CD3+ T cell interaction with OCs in both humans and hu-BLT mice.

## 2. Materials and Methods

### 2.1. Reagents, Cytokines, and Antibodies

RPMI 1640 (Life Technologies, Carlsbad, CA, USA) medium supplemented with 10% fetal bovine serum (FBS (Life Technologies, Carlsbad, CA, USA)) was used to culture the immune cells derived from human and humanized-BLT (hu-BLT) mice. Alpha-MEM (Life Technologies, Carlsbad, CA, USA) medium supplemented with 10% FBS was used to culture OCs. Human M-CSF, monoclonal antibodies for NK cells (anti-CD16 mAbs) and human ELISA kits for IFN-γ were obtained from Biolegend, San Diego, CA, USA. RANKL was obtained from PeproTech, Cranbury, NJ, USA, and recombinant human IL-2 (rh-IL2) was obtained from Hoffman La Roche, Little Falls, NJ, USA. Monoclonal antibodies for T cell human anti-CD3/CD28 mAbs were purchased from Stem Cell Technologies, Kingsway, Vancouver, BC, Canada.

### 2.2. Human NK Cells, T Cells, and Monocytes Isolation from PBMCs

All procedures involving healthy human peripheral blood were performed according to UCLA Institutional Review Board (IRB#11-000781 approved guidelines, and UCLA-IRB approved written informed consent was obtained from blood donors. We have previously demonstrated the process of isolating peripheral blood mononuclear cells (PBMCs) from peripheral blood using Ficoll-hypaque centrifugation. In this study, PBMCs were used to isolate NK cells, CD3+ T cells, CD4+ T cells, CD8+ T cells, γδ T cells, and monocytes using the EasySep^®^ human NK cell, EasySep^®^ human CD3+ T cell, EasySep^®^ human CD4 T, EasySep^®^ human CD8 T cell, EasySep^®^ human γδ T cell, and EasySep^®^ human monocyte enrichment kits, respectively, purchased from Stem Cell Technologies, Kingsway, Vancouver, BC, Canada. Cell-specific antibodies were used to measure the cell purity of each subset using flow cytometric analysis.

### 2.3. Cell Isolation of hu-BLT Mice

Procedures involving NSG and humanized-BLT (hu-BLT) mice were performed according to guidelines approved by UCLA Animal Research Committee (ARC 1997-136) in accordance with all federal, state, and local guidelines. Hu-BLT mice were prepared using NSG background as previously described [79,81]. For bone marrow cells, femurs were cut at both ends, and the single-cell suspension was obtained by flushing RPMI 1640 medium through femurs BM cells were then filtered using a 40 µm cell strainer. EasySep^®^ human monocyte enrichments kits purchased from Stem Cell Technologies, Kingsway, Vancouver, BC, Canada were used to isolate monocytes from bone marrow cells. The spleens were minced, and recovered splenocytes were filtered using a 40 µm cell strainer. Cells were then centrifuged at 4 °C for 5 min at 1500 rpm. ACK buffer was used to clear out red blood cells. Isolated splenocytes were then resuspended in RPMI medium and were centrifuged at 4 °C for 5 min at 1500 rpm. NK and T cells were isolated from splenocytes using EasySep^®^ human NK and T cell enrichment kits, respectively, obtained from Stem Cell Technologies, Kingsway, Vancouver, BC, Canada.

### 2.4. Human and hu-BLT Mice Osteoclasts (OCs) Generation

For OC generation, alpha-MEM medium (replenished every three days) was used to culture monocytes. On day 0 and day 3 of culture, the alpha-MEM medium was supplemented with M-CSF (25 ng/mL). On day 6 onwards of culture, the alpha-MEM medium was supplemented with M-CSF (25 ng/mL) and RANKL (25 ng/mL), after which, the medium was refreshed every 3 days using alpha-MEM medium supplemented with M-CSF (25 ng/mL) and RANKL (25 ng/mL). Multinucleate morphology of OCs was tested with TRAP staining [51]. Differentiation of OCs was verified using surface markers CD54, MHC-class I, MHC-class II, CD54, CD44, CD14, CD11b, B7H1, CD33, CD15, and CD124.

### 2.5. Cell Cultures

RPMI 1640 medium containing 10% fetal bovine serum (FBS) was used to culture NK and T cells and also their co-culture with OCs and sAJ2. NK cells from human and hu-BLT mice were activated with a combination of rh-IL-2 (1000 U/mL) and anti-CD16 mAbs (3 µg/mL) overnight and were then co-cultured with OCs and sAJ2 (OCs:NK:sAJ2; 1:2:4). Human γδ T and CD3+ T cells and hu-BLT mice CD3+ T cells were activated with rh-IL-2 (100 U/mL) and anti-CD3 (1µg/mL)/anti-CD28 mAbs (3 µg/mL) overnight, and cells were then cultured with OCs and sAJ2 (OCs:NK/T:sAJ2). Probiotic bacterium AJ2 is a combination of eight different strains of gram-positive probiotic bacteria (*Streptococcus thermophiles*, *Bifidobacterium longum*, *Bifidobacterium breve*, *Bifidobacterium infantis*, *Lactobacillus acidophilus*, *Lactobacillus plantarum*, *Lactobacillus casei*, and *Lactobacillus bulgaricus*) elected for their superior ability to activate NK cells. On days 2, 6, and 9, the cells were counted, and the supernatants from cultures were harvested for ELISA. Every supernatant collection day, the medium was refreshed with RPMI medium supplemented with rh-IL-2 (1500 U/mL for NK cells and 150 U/mL for γδ T cells and CD3+ T cells). On day 12, cells were counted, and cultures were terminated.

### 2.6. Enzyme-Linked Immunosorbent Assays (ELISAs)

IFN-γ ELISAs were performed as demonstrated previously. Procedure was performed according to manufacturer’s instructions. The concentration of IFN-γ was analyzed and obtained, and the standard curve was generated by either two- or three-fold dilution of IFN-γ recombinant available in the kit.

### 2.7. ^51^Chromium Release Cytotoxicity Assay

The chromium-51 (^51^Cr) release cytotoxicity assay was performed by incubating effector cells and ^51^Cr–labeled target cells for four hours at different effector to target ratios (5:1; 2.5:1. 1.25:1; 0.625:1). The supernatants were harvested from each sample after four hours of incubation, and the released radioactivity was counted using a gamma counter. The percentage specific cytotoxicity was calculated as follows:
$$\%\;\mathrm{Cytotoxicity}\;=\;\frac{\mathrm{Experimental}\;\mathrm{cpm}\;-\;\mathrm{spontaneous}\;\mathrm{cpm}}{\mathrm{Total}\;\mathrm{cpm}\;-\;\mathrm{spontaneous}\;\mathrm{cpm}}$$


Lytic units (LU) 30/10^6^ are calculated by using the inverse of the number of effector cells needed to lyse 30% of tumor cells × 100.

### 2.8. Statistical Analyses

All statistical analyses were performed using the GraphPad Prism 9 software. Different groups were compared using one-way ANOVA with a Bonferroni post-test. For in vitro studies, assessments were conducted using duplicate or triplicate samples. The number of stars represent the levels of statistical significance within those samples: **** (*p* value <0.0001), *** (*p* value 0.0001–0.001), ** (*p* value 0.001–0.01), * (*p* value 0.01–0.05).

## 3. Results

### 3.1. T Cell Subsets Were Unable to Target Oral Tumor Cancer Stems When Compared to NK Cells

The cytotoxic function of NK and T cells was compared using either untreated or IL-2-treated NK, CD3+ T, CD4+ T, CD8+ T, and γδ T cells as effectors against oral squamous carcinoma stem cells (OSCSCs) in ^51^Cr release cytotoxicity assay. OSCSCs are stem-like oral tumors. We have previously characterized these cells fully regarding their susceptibility to NK cell-mediated cytotoxicity in many previous publications. In this study, we used OSCSCs as the target to test and compare NK cell- and T cell-mediated cytotoxicity against these tumors. The cytotoxic function of T cell subsets was significantly lower when compared to NK cells, indicating the specificity of NK cells in targeting such tumors (Figure 1).

### 3.2. Osteoclasts and Probiotic Bacteria sAJ2 Induced Significant Cell Expansion in NK and T Cells

Our laboratory has previously demonstrated that OCs are capable of inducing expansion and functional activation of NK cells. In this study, human peripheral blood-derived NK cells were activated with rh-IL-2 (1000 U/mL) and anti-CD16 mAbs (3 µg/mL), and γδ T cells and CD3+ T cells were activated with rh-IL-2 (100 U/mL) and anti-CD3+ (1 µg/mL)/anti-CD28 mAbs (3 µg/mL) overnight. Cells were then cultured with OCs and probiotic bacteria sAJ2 (1:2:4; OCs:NK/T:sAJ2). We have previously reported that during supercharging of NK and T cells, OCs were eliminated by day 3 to day 6, which we have determined using confocal microscopy and imaging [52]. In the current study, the first time point to assess the number of NK and T cells was day 6, on which we counted only NK and T cells and no OCs. Microscopically, we could not observe any large multinucleated OCs in the cultures of NK or T cells with OCs at this time point on. We observed that the combination of OCs and probiotic treatment induced a significantly higher rate of cell expansion in NK and γδ T cells but lower in CD3+ T cells in comparison to controls (Figure 2).

### 3.3. Osteoclasts Induced Functional Activation of NK and T Cells

We assessed the OCs and sAJ2-induced effect on the cytotoxic function of NK and T cells against OSCSCs. The combination of OCs and sAJ2 induced a significantly higher increase in cytotoxic activity of NK cells compared to γδ T or CD3+ T cells (Figure 3A). The combination of OCs and sAJ2 probiotic treatment resulted in significantly increased levels of IFN-γ secretion in NK cells, γδ T, and CD3+ T cells in comparison to controls (Figure 3B,C). The highest increase of IFN-γ secretion was observed in CD3+ T cells both on the overall levels (Figure 3B,C) as well as in per 1 million cells (Figure 4).

### 3.4. Like Human OCs, hu-BLT Mice Derived OCs also Induced Increased Numbers and Functional Activation in NK and T Cells

Many mouse models were used to study immune cell interaction, but mostly differences in human and mouse immune systems were observed [75,76,82,83]. To date, the humanized-BLT mouse (hu-BLT) is found to be the most representative mouse model to study human pathologies or to assess immune function [77,78,79,80]. Similar to human immune cells, hu-BLT mice-derived NK and CD3+ T cells were activated overnight. Cells were then cultured with OCs and probiotic bacteria sAJ2 (1:2:4; OCs:NK/T:sAJ2). The combination of OCs and probiotic treatment induced a significantly higher increase in the number of NK and CD3+ T cells in comparison to controls (Figure 5A). Similarly, IFN-γ secretions from both NK and CD3+ T cells of hu-BLT mice increased when treated with OCs and sAJ2 probiotic bacteria (Figure 5B). Accordingly, IFN-γ secretions per million cells were also elevated in OC and sAJ2 probiotic bacteria-treated NK and T cells, albeit NK cells had some increased levels when compared to those obtained from T cells (Figure 5C).

## 4. Discussion

One of the major limiting factors for successful cell therapy is not having sufficient counts of functionally competent NK cells with the existing cell therapeutic strategies. In addition, existing NK cell therapeutics either do not yield enough cells, or the infused cells become inactivated by the tumor microenvironment and, therefore, have a short life span [84,85,86,87,88]. To counter both factors, we developed NK cell therapeutics that expand substantially and are not inactivated by the tumor microenvironment using osteoclasts (OCs) as feeder cells (manuscript submitted). We previously have introduced OCs as major activators of NK cells in numerous publications [52,53,74,89]. Not only do they exhibit the potential for inducing cell expansion in NK cells, unlike many other activators of NK cells, but they also have a great ability to increase granule release and augmented secretion of TNF-a and TRAIL, leading to increased cytotoxicity in short-term as well as long-term periods, respectively [52,90,91,92]. In addition to OCs, probiotic bacteria sAJ2 were used for NK cell expansion; sAJ2 was formulated specifically for NK cell activation [10]. Probiotics are well-known NK cell activators, and they mediate their effect partly by upregulation of IFN-γ secretion in NK cells. Since we used sonicated AJ2 probiotic bacteria to activate the NK cells, many different components and bacterial metabolites were likely the activators of NK cells. AJ2 was found to be the best activator of NK cells, and it induced the highest levels of IFN-γ when compared to many other activators of NK cells. In addition, it imparted survival advantage to NK cells (Figure 6). AJ2 probiotic bacteria along with rh-IL-2 highly upregulated secretion levels of cytokines and augmented the cytotoxic activity in NK cells, whereas they had lower levels of activation when added to T cells. Therefore, there is a substantial difference between the ability of AJ2 bacteria in activating NK cells vs. T cells. At the present time, it is unclear how these differences affect the function of NK and T cells.

Expanded NK cells by OCs and sAJ2 were named supercharged NK cells (sNK) because of their highly functional attributes as well as their significant expansion (Figure 6). In in vivo experiments, one injection of sNK cells was capable of substantially decreasing the growth of tumor cells in pancreatic [53], oral [54], and melanoma (manuscript in prep.) models. It is, however, unclear whether the effect of OCs on NK cells is specific or whether other cytotoxic immune cells can also be expanded and activated by osteoclasts. We chose to focus on γδ T cells and pan T cells, which also include CD8+ T cells. Indeed, in our previous publications we established that sNK cells have the capability of selecting and proliferating T cells [74].

In terminal cancer, patients suffer from inadequate numbers and functions of both NK and T cells; therefore, supplementation of competent NK and T cells would likely provide competent cells with a great ability to target tumor cells. To accomplish this, we hypothesized that similarly to NK cells, γδ T cells and CD3+ T cells will also respond to osteoclasts and expand and become functionally activated. Cultures of γδ T cells in the presence of OCs and sAJ2 probiotic bacteria resulted in a greater expansion, and the levels decreased at day 9 and 12 when compared to NK and CD3+ T cells. Thus, it appears that γδ T cells expand faster than NK and CD3+ T cells at the early time points, but then they contract earlier than NK and CD3+ T cells (Figure 2). CD3+ T cells upon activation with rh-IL-2 and anti-CD3/28 mAbs expand substantially, and OCs increase the expansion. However, since the background of CD3+ T cells without OCs is higher, the differences are not significant on days 6 and 9, whereas on day 12 significant differences were noted. In general, for all three immune subsets we saw a significant expansion when these cells were cultured in the presence of OCs and sAJ2 probiotic bacteria.

Unlike NK cells, CD3+ T cells or γδ T cells when activated were not capable of killing NK cell-specific targets. Accordingly, OC-expanded CD3+ T cells or γδ T cells were also not capable of lysing NK cell targets, even though they are highly activated functionally. These results indicate the specific nature of NK targets in NK cell-mediated cytotoxicity (Figure 3A). Similar to NK cells, osteoclast-expanded CD3+ T or γδ T cells exhibit higher potential to secrete IFN-γ. γδ T cells secreted significant levels of IFN-γ from days 2–9, and at day 12 no significant differences could be seen, whereas both NK and CD3+ T cells continued secreting significant levels at all time points tested (Figure 3B). However, when overall levels of IFN-γ secretion were evaluated for the three subsets, they all had significant levels of IFN-γ secretion, albeit it appears the γδ T cells have slightly lower levels on average in comparison to NK and CD3+ T cells (Figure 3C). When assessing IFN-γ release per cell basis, γδ T cells significantly released IFN-γ on days 6 and 9 but not on day 12, whereas CD3+ T cells released significant levels of IFN-γ at all time points (Figure 4). Overall, it appears that on average, CD3+ T cells had highest upregulation in IFN-γ secretion, second were NK cells, and the lowest levels were seen in γδ T cells when cultured with OCs and sAJ2 probiotic bacteria.

As a mouse model for human disease, we used hu-BLT mice in order to determine the safety and efficacy of sNK cells in targeting several kinds of human cancers [53,54]. For the in vivo anti-tumor assay, we harvested tumors (oral, melanoma, or pancreatic) from hu-BLT mice to compare the tumor size, weight, and cell-counts among sNK cell-treated group vs. no treatment. sNK cell treatment resulted in a significantly smaller tumor mass when compared to the no-treatment group of hu-BLT tumor-bearing mice. Human immune cell infiltration was measured in hu-BLT mice-derived tumors. Tumors harvested from the sNK cell treatment group expressed about 3–9-fold increase in human CD45+ infiltrating immune cells when compared to tumors harvested from the no-treatment group [53,54]. Reduced levels of tumor growth rate and significantly higher levels of IFN-γ secretion were observed in the tumor cultures of the sNK-treated group in comparison to the no-treatment group [53]. A differentiated tumor phenotype was found in tumors from the sNK cells treatment group. Tumors harvested from sNK-treated hu-BLT mice were highly positive for MHC-class I, CD54, and B7H1 surface expression levels and were resistance to NK cell-induced cytotoxicity [53]. Increased percentages of CD3+CD8+ T cells were observed in bone marrow, splenocytes, and PBMCs of sNK cell-treated hu-BLT mice [53,74]. In addition, sNK cell treatment reverted the cytotoxic activity and levels of cytokine secretion in immune cells from tissue compartments of tumor-bearing hu-BLT mice. These observations demonstrate the therapeutic benefits of sNK cells as cancer immunotherapy.

In the current study, we also determined whether NK and T cell activation through OCs is evident in hu-BLT mice. Similarly to humans, culture of either hu-BLT mice-derived NK or T cells with OCs exhibited increased cell expansion and significantly higher levels of IFN-γ secretion. We were unable to assess γδ T cells in this study due to the low number of isolated cells. These experiments indicated the close similarity between humans and humanized mice in supercharging the NK and CD3+ T cells.

## 5. Conclusions

In conclusion, this study indicates that γδ T or CD3+ T cells can be expanded and supercharged like NK cells; therefore, studies can be designed to determine whether a combination of such effectors together would be more effective in targeting and eliminating the tumors than each subset alone.

## Figures and Tables

**Figure 1 cells-13-00213-f001:**
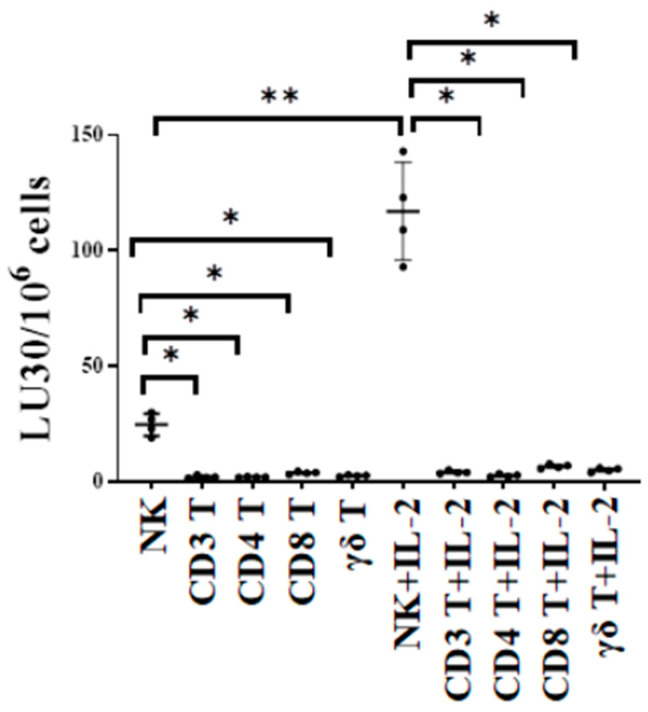
NK cells killed higher number of oral cancer stem cells compared to T cells. Healthy human-donor-derived NK, CD3+ T, CD4+ T, CD8+ T, and γδ T cells were either left untreated or were treated with rh-IL-2 (NK: 1000 U/mL, and T cells:100 U/mL) overnight and were then used as effectors against OSCSCs in standard 4 h ^51^Cr release cytotoxicity assay. The lytic units (LU) 30/10^6^ cells were determined using the inverse number of effector cells required to lyse 30% of OSCSCs × 100 (*n* = 4). ** (*p* value 0.001–0.01), * (*p* value 0.01–0.05).

**Figure 2 cells-13-00213-f002:**
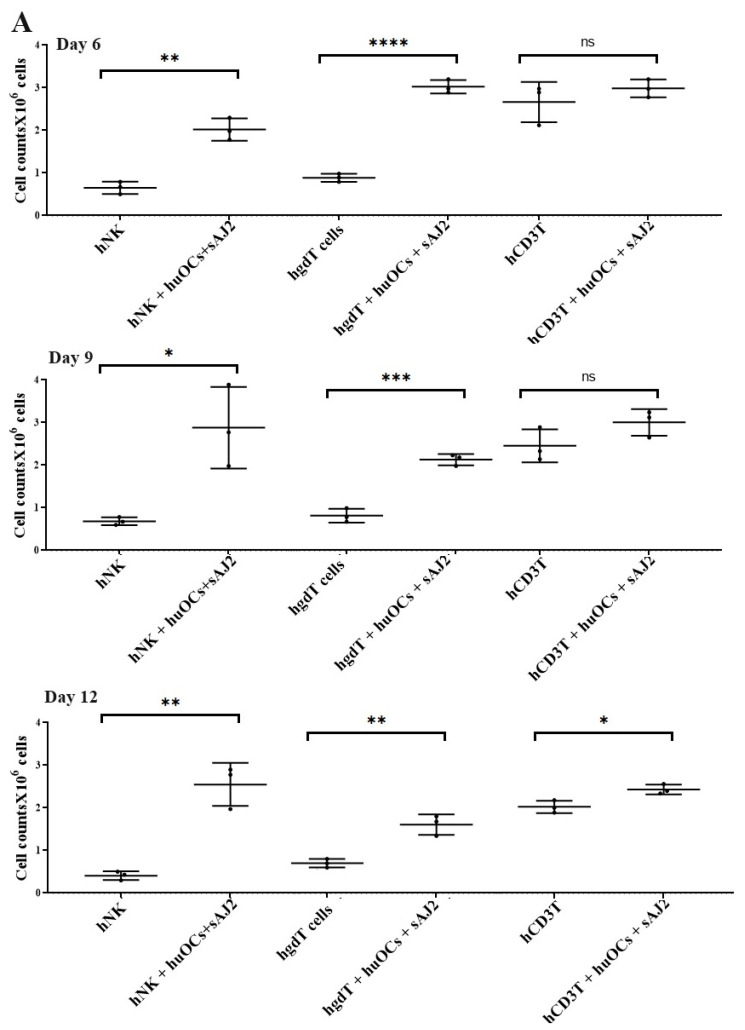
Cell counts of human NK and T cells when they were co-cultured with OCs and sAJ2 probiotic bacteria. Human monocytes, NK cells, γδ T cells, and CD3+ T cells were isolated from PBMCs. OCs were generated as described in the Materials and Methods section. NK cells (1 × 10^6^ cells/mL) were treated with a combination of rh-IL-2 (1000 U/mL) and anti-CD16 mAbs (3 µg/mL) overnight. Cells were then cultured with OCs and sAJ2 (1:2:4; OCs:NK:sAJ2). γδ T and CD3+ T cells were treated with rh-IL-2 (100 U/mL) and anti-CD3+ (1µg/mL)/anti-CD28 (3 µg/mL) for 18–20 h. Cells were then cultured with OCs and sAJ2 (1:2:4; OCs:γδT or CD3 + T:sAJ2). Cells were counted on days 6, 9, and 12 from the co-cultures (*n* = 3) (**A**). Cumulative cell counts are shown in Figure (*n* = 9) (**B**). *n* is number of healthy donors. **** (*p* value <0.0001), *** (*p* value 0.0001–0.001), ** (*p* value 0.001–0.01), * (*p* value 0.01–0.05).

**Figure 3 cells-13-00213-f003:**
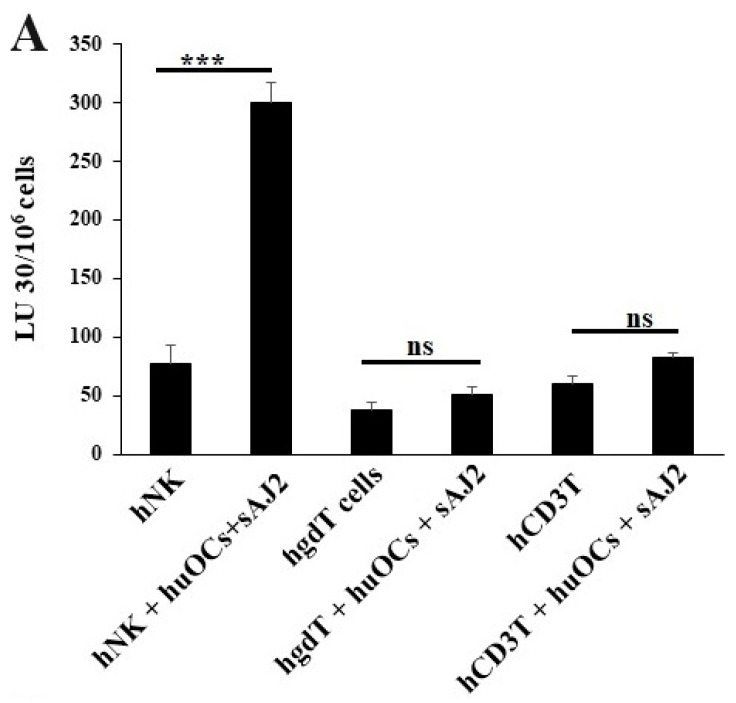
Functional activation of human NK and T cells when they were co-cultured with OCs and sAJ2 probiotic bacteria. Cells were cultured as described in Figure 2. NK, γδT, or CD3+ T cell-mediated cytotoxicity against OSCSCs was determined on day 12 of co-culture using a standard 4 h ^51^Cr release cytotoxicity assay. The lytic units 30/10^6^ cells were determined using the inverse number of NK, γδT or CD3+ T cells required to lyse 30% of OSCSCs × 100. Average and std dev. of two experiments are shown in figure (**A**). Cells were cultured as described in Figure 2. The supernatants were harvested on days 2, 6, 9, and 12 from the co-cultures to determine the IFN-γ secretion level using single ELISA (*n* = 4) (**B**). Cumulative IFN-γ secretion is shown in Figure (*n* = 16) (**C**). *n* is number of healthy donors **** (*p* value <0.0001), *** (*p* value 0.0001–0.001), ** (*p* value 0.001–0.01), * (*p* value 0.01–0.05).

**Figure 4 cells-13-00213-f004:**
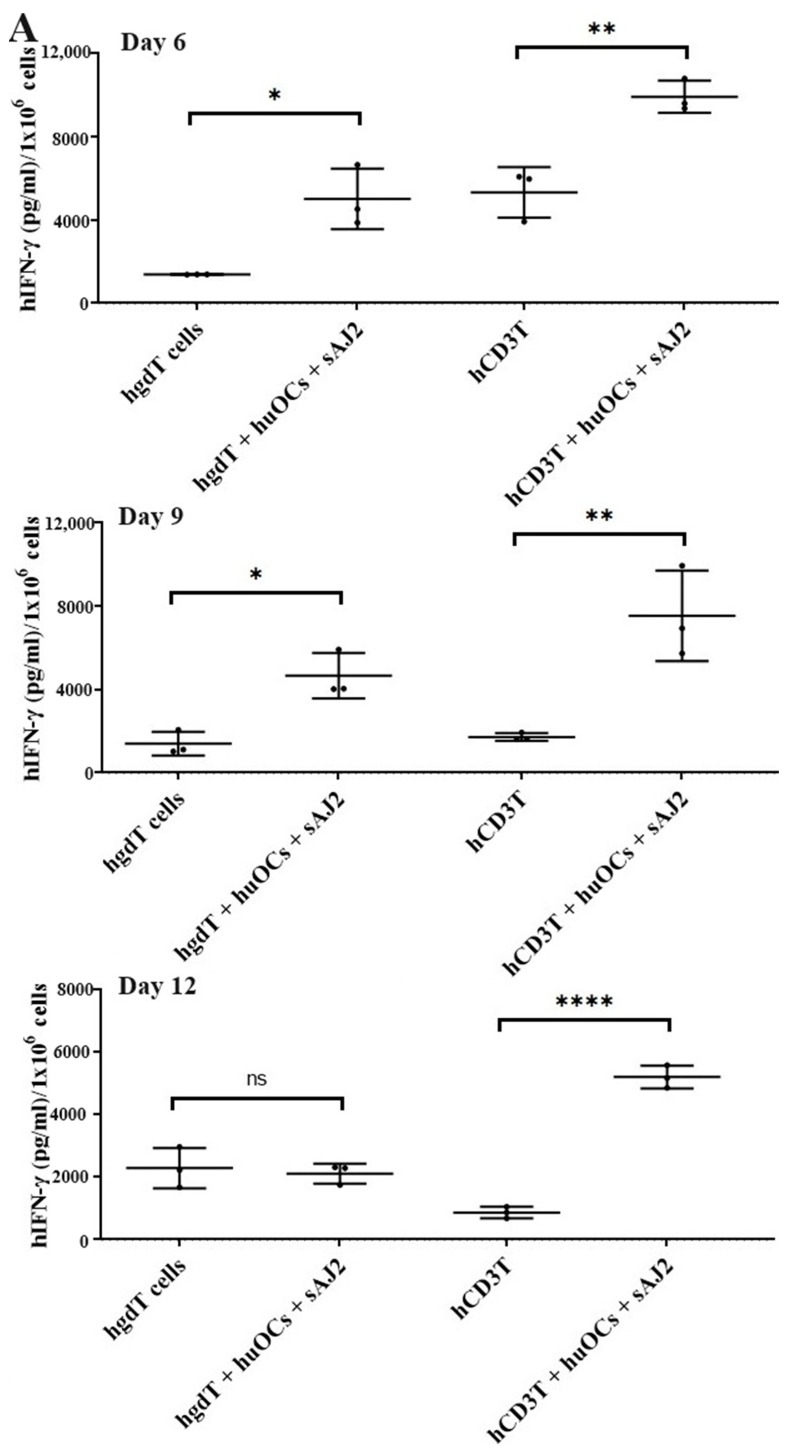
IFN-γ per million human NK and T cells when they were co-cultured with OCs and sAJ2 probiotic bacteria. Cells were cultured as described in Figure 2. Cells were counted and, the supernatants were harvested on days 6, 9, and 12 from the co-cultures to determine the IFN-γ secretion level using single ELISA. The IFN-γ secretion level was determined per million cells (*n* = 3) (**A**). Cumulative IFN-γ secretion per million cell counts is shown in Figure (*n* = 9) (**B**). *n* is number of healthy donors. **** (*p* value <0.0001), *** (*p* value 0.0001–0.001), ** (*p* value 0.001–0.01), * (*p* value 0.01–0.05).

**Figure 5 cells-13-00213-f005:**
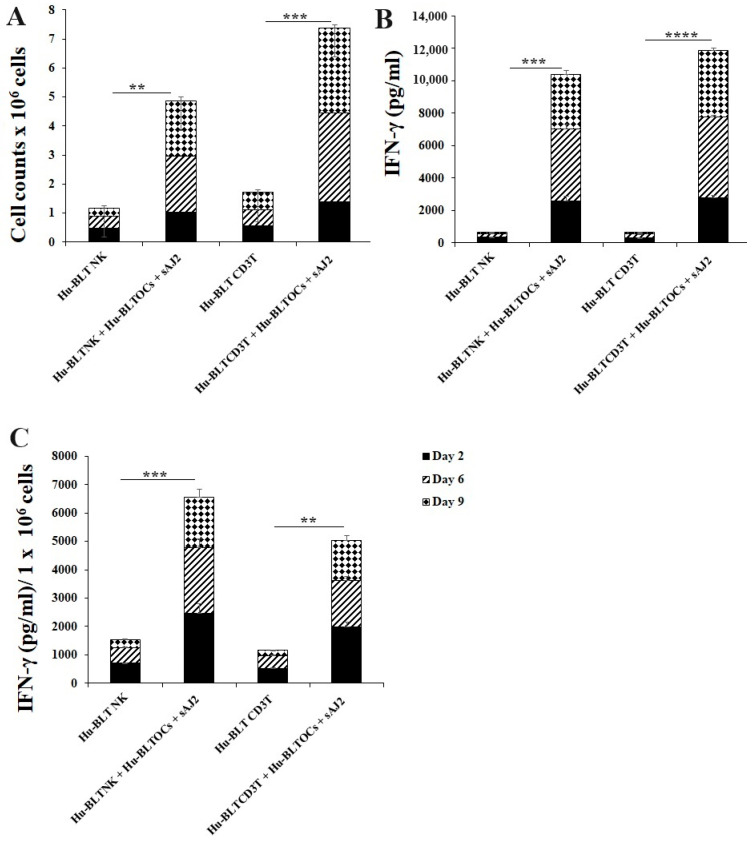
IFN-γ of hu-BLT NK and T cells when cultured with OCs and sAJ2. Hu-BLT mice NK and CD3 + T cells were isolated from splenocytes, and hu-BLT mice monocytes were isolated from bone marrow as described in the Material and Methods section. OCs were generated as described in the Materials and Methods section. NK cells (1 × 10^6^ cells/mL) were treated with a combination of IL-2 (1000 U/mL) and anti-CD16 mAbs (3 µg/mL) overnight. Cells were then cultured with OCs and sAJ2 (1:2:4; OCs:NK:sAJ2). CD3+ T cells were treated with rh-IL-2 (100 U/mL) and anti-CD3+ (1µg/mL)/anti-CD28 (3 µg/mL) for 18–20 h. Cells were then cultured with OCs and sAJ2 (1:2:4; OCs: CD3 + T:sAJ2). Cells were counted (**A**), and the supernatants were harvested on days 6, 9, and 12 from the co-cultures to determine the IFN-γ secretion level using single ELISA (**B**), which was determined per million of cells (**C**). The average of the two experiments is shown in the figure. **** (*p* value <0.0001), *** (*p* value 0.0001–0.001), ** (*p* value 0.001–0.01).

**Figure 6 cells-13-00213-f006:**
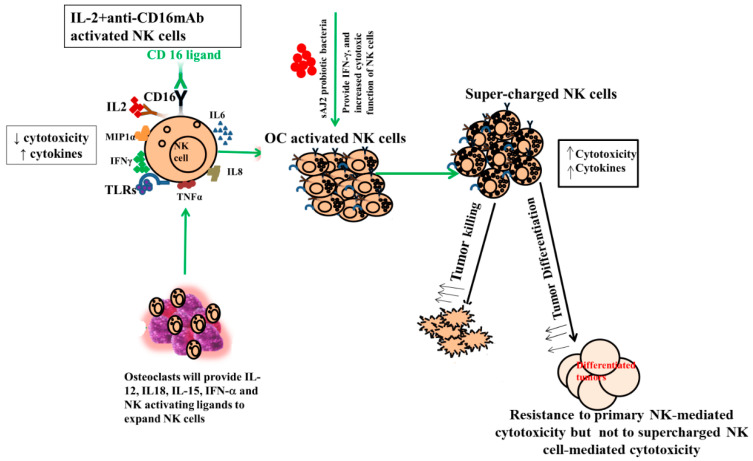
Role of OCs and probiotic bacteria in supercharging NK cells. Supercharged NK cells secrete higher cytokines and are highly cytotoxic in comparison to primary NK cells. OCs were generated as described in the Materials and Methods section. NK cells (1 × 10^6^ cells/1ml) were activated with rh-IL-2 (1000 U/mL) and anti-CD16 mAbs (3 μg/mL) overnight. Cells were then co-cultured with OCs and were treated with sAJ2 probiotic bacteria (1:2:4:OCs:NK:sAJ2). The medium was refreshed every three days for an average of 27–36 days. Supercharged NK cells were analyzed for their cytokine secretion function and cytotoxicity against CSCs.

## Data Availability

Data generated or analyzed during the study are included in this submitted article.

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
