# Peer review of "Osteoclasts and Probiotics Mediate Significant Expansion, Functional Activation and Supercharging in NK, γδ T, and CD3+ T Cells: Use in Cancer Immunotherapy"

_cells, 2024, doi:10.3390/cells13030213_

Round 1

Reviewer 1 Report

Comments and Suggestions for Authors

The authors have submitted an original manuscript entitled Osteoclasts mediate significant expansion, functional activation and supercharging of NK, CD3+ T and γδ T cells; use in cancer immunotherapy.

The manuscript is precise and reasonably written. To improve the manuscript, technical corrections and additions to the text part are needed to clarify the importance of the joint action of osteoclasts and probiotic bacteria. In particular, the title of the manuscript needs to be changed and the term probiotic bacteria needs to be included in the title, as it is important for explaining the full results. In addition, it would be desirable to add the markers analysed to verify osteoclast differentiation in part M&M 2.4.

For a clear presentation of the importance of probiotic bacteria for the observed effects and effects on the activation of different classes of lymphocytes, the proposed mechanisms (direct or indirect through bacterial metabolites) should be indicated in the discussion.

An additional figure showing the putative mechanisms of OC and sAJ2 would greatly enhance the visual presentation of the manuscript. I suggest accepting this manuscript.

Minor corrections

line 40: that

line 60: Osteoclasts (OCs)

line 68: peripheral blood mononuclear cells (PBMCs)

sAJ2 probiotic bacteria

Comments on the Quality of English Language

The authors have submitted an original manuscript entitled Osteoclasts mediate significant expansion, functional activation and supercharging of NK, CD3+ T and γδ T cells; use in cancer immunotherapy.

The manuscript is precise and reasonably written. To improve the manuscript, technical corrections and additions to the text part are needed to clarify the importance of the joint action of osteoclasts and probiotic bacteria. In particular, the title of the manuscript needs to be changed and the term probiotic bacteria needs to be included in the title, as it is important for explaining the full results. In addition, it would be desirable to add the markers analysed to verify osteoclast differentiation in part M&M 2.4.

For a clear presentation of the importance of probiotic bacteria for the observed effects and effects on the activation of different classes of lymphocytes, the proposed mechanisms (direct or indirect through bacterial metabolites) should be indicated in the discussion.

An additional figure showing the putative mechanisms of OC and sAJ2 would greatly enhance the visual presentation of the manuscript. I suggest accepting this manuscript.

Minor corrections

line 40: that

line 60: Osteoclasts (OCs)

line 68: peripheral blood mononuclear cells (PBMCs)

sAJ2 probiotic bacteria

Author Response

We thank the reviewer for his/her insights. Here please find our response:

To improve the manuscript, technical corrections and additions to the text part are needed to clarify the importance of the joint action of osteoclasts and probiotic bacteria.

Response: We have previously compared the role of OCs in comparison to monocytes, macrophages, dendritic cells, K562, OSCSCs, MP2, and PBMCs, and found OCs to be potent activators of NK cells [1-3]. OC-mediated effect in the induction of cell expansion, increase cytotoxicity and secretion of cytokines and chemokines by NK cells was much higher than any other tested feeder cells [1-3]. OCs secrete the majority of cytokines required for the activation of NK cells such as IL-15, IL-12, IL-18 and IFN-α, and express lower levels of MHC-class I and II, CD14, CD11b and CD54 on the surface, and minimally upregulate MHC-class I surface expression when treated with supernatants from the activated NK cells  (NK cell supernatant is known to increase MHC-class I expression due to increased levels of IFN-g secretion) [1]. To sum up, the underlying mechanisms by which OCs are able to expand functionally potent NK cells is the lower expression of MHC-class I together with increased release of IL-15, IL-12, IL-18 and IFN-α. In addition, OCs also exhibit higher expression of NKG2D ligands which are known to increase NK function [1]. Probiotics sAJ2 also contribute to increased cytokine secretion by NK cells specially IFN-γ which could facilitate signals required for NK cell expansion. In addition, probiotic bacteria increased the survivability of NK cells [3, 4]. Therefore, combination of both probiotics and OCs induced signals which participate additively in the expansion and functional activation of NK cells. OC and probiotic-expanded super-charged NK cells are highly potent in targeting CSCs, and in the differentiation of CSCs both in in-vitro and in-vivo studies [5-9].

In particular, the title of the manuscript needs to be changed and the term probiotic bacteria needs to be included in the title, as it is important for explaining the full results.

Response: We have now updated the title as ’’ Osteoclasts and probiotics mediate significant expansion, functional activation and supercharging of NK, CD3+ T and γδ T cells; use in cancer immunotherapy”

In addition, it would be desirable to add the markers analysed to verify osteoclast differentiation in part M&M 2.4.

Response: We have now added the information in M&M 2.4 ’’Multinucleate morphology of OCs was tested with TRAP staining [1]. Differentiation of OCs was verified using surface markers CD54, MHC-class I, MHC-class II, CD54, CD44, CD14, CD11b, B7H1, CD33, CD15, and CD124 [1].’’

For a clear presentation of the importance of probiotic bacteria for the observed effects and effects on the activation of different classes of lymphocytes, the proposed mechanisms (direct or indirect through bacterial metabolites) should be indicated in the discussion.

Response: We have added a paragraph to the discussion.

Probiotic bacteria sAJ2  was used in combination with OCs for NK cell expansion. sAJ2 was formulated specifically to increase the function of NK cells [4]. Probiotics are well-known activators of NK cells, and they mediate their effect partly by increasing IFN-γ secretion by the NK cells [4]. Since we used sonicated AJ2 probiotic bacteria to activate the NK cells, many different components and bacterial metabolites were likely the activators of NK cells. AJ2 was found to be the best activator of NK cells and it induced the highest levels of IFN-γ when compared to many other activators of NK cells [4]. In addition, it imparted survival advantage to NK cells [3, 4]. AJ2 probiotic bacteria in combination with IL-2 triggered significant secretion of cytokines  and augmented the cytotoxicity of NK cells, whereas they had lower levels of activation when added to T cells [10]. Therefore, there is a substantial difference between the ability of AJ2 bacteria in activating NK cells vs. T cells. At the moment it is not clear how these differences effect the function of NK and T cells.

An additional figure showing the putative mechanisms of OC and sAJ2 would greatly enhance the visual presentation of the manuscript. I suggest accepting this manuscript.

Response: We have now added the figure showing putataive mechanism of OCs and sAJ2

Minor corrections

line 40: that

Response: Corrected

line 60: Osteoclasts (OCs)

Response: Corrected

line 68: peripheral blood mononuclear cells (PBMCs)

Response: Corrected

sAJ2 probiotic bacteria

Response: Corrected

  1. Tseng, H.C., et al., Bisphosphonate-induced differential modulation of immune cell function in gingiva and bone marrow in vivo: role in osteoclast-mediated NK cell activation. Oncotarget, 2015. 6(24): p. 20002-25.
  2. Kaur, K., et al., Sequential therapy with supercharged NK cells with either chemotherapy drug cisplatin or anti-PD-1 antibody decreases the tumor size and significantly enhances the NK function in Hu-BLT mice. Frontiers in Immunology, 2023. 14.
  3. Kaur, K., et al., Novel Strategy to Expand Super-Charged NK Cells with Significant Potential to Lyse and Differentiate Cancer Stem Cells: Differences in NK Expansion and Function between Healthy and Cancer Patients. Frontiers in Immunology, 2017. 8.
  4. Bui, V.T., et al., Augmented IFN-γ and TNF-α Induced by Probiotic Bacteria in NK Cells Mediate Differentiation of Stem-Like Tumors Leading to Inhibition of Tumor Growth and Reduction in Inflammatory Cytokine Release; Regulation by IL-10. Frontiers in Immunology, 2015. 6.
  5. Kaur, K., et al., Novel Strategy to Expand Super-Charged NK Cells with Significant Potential to Lyse and Differentiate Cancer Stem Cells: Differences in NK Expansion and Function between Healthy and Cancer Patients. Front Immunol, 2017. 8: p. 297.
  6. Bui, V.T., et al., Augmented IFN-γ and TNF-α Induced by Probiotic Bacteria in NK Cells Mediate Differentiation of Stem-Like Tumors Leading to Inhibition of Tumor Growth and Reduction in Inflammatory Cytokine Release; Regulation by IL-10. Front Immunol, 2015. 6: p. 576.
  7. Kaur, K., et al., Probiotic-Treated Super-Charged NK Cells Efficiently Clear Poorly Differentiated Pancreatic Tumors in Hu-BLT Mice. Cancers (Basel), 2019. 12(1).
  8. Kaur, K., et al., Super-charged NK cells inhibit growth and progression of stem-like/poorly differentiated oral tumors in vivo in humanized BLT mice; effect on tumor differentiation and response to chemotherapeutic drugs. Oncoimmunology, 2018. 7(5): p. e1426518.
  9. Dong, H., I. Rowland, and P. Yaqoob, Comparative effects of six probiotic strains on immune function in vitro. British Journal of Nutrition, 2012. 108(3): p. 459-470.
  10. Kaur, K., et al., Osteoclast-expanded super-charged NK-cells preferentially select and expand CD8+ T cells. Scientific Reports, 2020. 10(1): p. 20363.

Reviewer 2 Report

Comments and Suggestions for Authors

This paper was difficult to follow due to the quality of the English language. Also, the graphs were rather small and should either be broken up or enlarged for better viewing. On lines 13 and 274, I am assuming the authors meant TRAIL and not TRIAL for the cytokines OCs make. Also, how were NK and T cells specifically counted in the co-culture system to not include the added OCs? This should be indicated in the materials and methods.

Comments on the Quality of English Language

Overall, improvement in the quality of the English language is recommended. Specific points of certain sentences were hard to determine, so there is an overall lack of understanding around the significance of the results and this paper. 

Author Response

We thank the reviewer for his/her insights. Here please find our response

This paper was difficult to follow due to the quality of the English language.

Response: We have now edited the English language of the manuscript carefully.

Also, the graphs were rather small and should either be broken up or enlarged for better viewing.

Response: We have now broken the graphs for better view. Thank you

On lines 13 and 274, I am assuming the authors meant TRAIL and not TRIAL for the cytokines OCs make.

Response: Corrected

Also, how were NK and T cells specifically counted in the co-culture system to not include the added OCs? This should be indicated in the materials and methods.

Response: We have previously reported that during supercharging of NK and T cells, OCs were eliminated by day 3-day 6, which we have determined using confocal microscopy and imaging [1]. In current study, the first time point to assess the numbers of NK and T cells was day 6 which we counted only NK and T cells and no OCs.  Microscopically, we could not observe any large multinucleated OCs in the cultures of NK or T cells with OCs at this time point on.

  1. Kaur, K., et al., Sequential therapy with supercharged NK cells with either chemotherapy drug cisplatin or anti-PD-1 antibody decreases the tumor size and significantly enhances the NK function in Hu-BLT mice. Frontiers in Immunology, 2023. 14.

Reviewer 3 Report

Comments and Suggestions for Authors

Authors described about osteoclast activates NK cells.

1. Please, add the declaration of helsinki.

2. Please, explain the cancer stem like oral tumors in detail.

3. Please, explain functionally activated NK cells. What did you define? CD56 dim NK cell fraction?

4. Please, add the in vivo anti-tumor assay.

Author Response

We are thankful to the Reviewer for his/her insights. Here please find our response

  1. Please, add the declaration of helsinki.

Response: This study was performed using immune cells from healthy individuals at academic research institute (University of California, Los Angeles, CA, USA), and required the approval of the Institutional Review Board (IRB). We have added this information in section 2.2 ” All procedures involving healthy human peripheral blood were performed according to guidelines approved by the UCLA Institutional Review Board (IRB), and written informed consents approved by UCLA-IRB were obtained from healthy individuals”.

We have now added IRB number:  IRB#11-000781

  1. Please, explain the cancer stem like oral tumors in detail.

Response: Oral squamous carcinoma stem-like cells (OSCSCs) are the stem-like oral tumors. We have previously characterized these cells fully regarding their surface expression, markers of stemness and susceptibility to NK cell-mediated cytotoxicity in many previous publications [1-3]. In this study, we used OSCSCs as targets to test and compare NK cell- and T cell-mediated cytotoxicity against these tumors.

  1. Please, explain functionally activated NK cells. What did you define? CD56 dim NK cell fraction?

Response: We used CD16highCD56dim NK cells which are cytotoxic fraction of NK cells. This subset constitutes 90% of total NK cells in the peripheral blood mononuclear cells (PBMCs) and play key role in NK cell-mediated cytotoxicity and ADCC function.

NK cells can be functionally activated by NK cell-specific cytokines such as IL-2, IL-12 and IL-15, or by NK cell specific antibodies such as anti-CD16 mAbs. In this study, NK cells were activated with a combination of IL-2 and anti-CD16mAb before they were cultured with OCs and sAJ2 probiotic bacteria. Activated NK cells play two main roles: 1. direct cytotoxicity and mediating ADCC; 2. induce differentiation of stem-like cells through secreted cytokines especially IFN-γ and TNF-α [1, 4].

  1. Please, add the in vivo anti-tumor assay.

Response: For in-vivo anti-tumor assay, we harvested tumors (oral, melanoma, or pancreatic) from hu-BLT mice and compared the tumor size, weight and cell-counts among supercharged NK cell treated group vs. no treatment [5, 6]. Supercharged NK cell treatment resulted in significantly smaller tumor mass in comparison to no treatment in hu-BLT tumor-bearing mice [5, 6]. Human immune cell infiltration was also measured in hu-BLT mice-derived tumors, approximately 3-9 fold higher human CD45+ infiltrating immune cells were found in tumors from the supercharged NK cell treatment group vs. no treatment group [5, 6].  When hu-BLT mice-derived tumors were cultured, growth rate was significantly lower and IFN-γ secretion levels were significantly higher in tumors from sNK treated group in comparison to no treatment group[5, 6]. Differentiated tumor phenotype was found in tumors from supercharged NK cells treatment group, tumors expressed higher surface levels of MHC-class I, CD54 and B7H1 and, were highly resistant to primary NK cell-mediated cytotoxicity [4-7]. Supercharged NK cells treatment resulted in increased proportions of CD3+CD8+T cells in bone marrow, spleen and peripheral blood of hu-BLT mice [5, 6, 8]. In addition, supercharged NK cell therapy restored the cytotoxic function and cytokine secretion of immune cells in spleen, peripheral blood, gingiva, pancreas, and bone-marrow of tumor-bearing hu-BLT mice [5, 6, 8]. These findings demonstrate the significance and efficacy of supercharged NK cells in treatment of cancer.

  1. Tseng, H.C., N. Cacalano, and A. Jewett, Split anergized Natural Killer cells halt inflammation by inducing stem cell differentiation, resistance to NK cell cytotoxicity and prevention of cytokine and chemokine secretion. Oncotarget, 2015. 6(11): p. 8947-59.
  2. Jewett, A., Y.G. Man, and H.C. Tseng, Dual functions of natural killer cells in selection and differentiation of stem cells; role in regulation of inflammation and regeneration of tissues. J Cancer, 2013. 4(1): p. 12-24.
  3. Bui, V.T., et al., Augmented IFN-γ and TNF-α Induced by Probiotic Bacteria in NK Cells Mediate Differentiation of Stem-Like Tumors Leading to Inhibition of Tumor Growth and Reduction in Inflammatory Cytokine Release; Regulation by IL-10. Front Immunol, 2015. 6: p. 576.
  4. Tseng, H.C., et al., Increased lysis of stem cells but not their differentiated cells by natural killer cells; de-differentiation or reprogramming activates NK cells. PLoS One, 2010. 5(7): p. e11590.
  5. Kaur, K., et al., Super-charged NK cells inhibit growth and progression of stem-like/poorly differentiated oral tumors in vivo in humanized BLT mice; effect on tumor differentiation and response to chemotherapeutic drugs. OncoImmunology, 2018. 7(5): p. e1426518.
  6. Kaur, K., et al., Probiotic-Treated Super-Charged NK Cells Efficiently Clear Poorly Differentiated Pancreatic Tumors in Hu-BLT Mice. Cancers, 2020. 12(1): p. 63.
  7. Kozlowska, A.K., et al., Differentiation by NK cells is a prerequisite for effective targeting of cancer stem cells/poorly differentiated tumors by chemopreventive and chemotherapeutic drugs. J Cancer, 2017. 8(4): p. 537-554.
  8. Kaur, K., et al., Osteoclast-expanded super-charged NK-cells preferentially select and expand CD8+ T cells. Scientific Reports, 2020. 10(1): p. 20363.

Round 2

Reviewer 3 Report

Comments and Suggestions for Authors

The manuscript is well revised.

Author Response

We thank the reviewer for his/her insights.